# Wave Glider Observations of Surface Waves During Three Tropical Cyclones in the South China Sea

**Di Tian** [1], **Han Zhang** [1,2,*], **Wenyan Zhang** [3], **Feng Zhou** [1,4,*], **Xiujun Sun** [5,6], **Ying Zhou** [7] **and Daoxun Ke** [1,8]

[1]   State Key Laboratory of Satellite Ocean Environment Dynamics, Second Institute of Oceanography, Ministry of Natural Resources, Hangzhou 310012, China; tiandi@sio.org.cn (D.T.); kedaoxun@sio.org.cn (D.K.)
[2]   Southern Marine Science and Engineering Guangdong Laboratory (Zhuhai), Zhuhai 519082, China
[3]   Institute of Coastal Research, Helmholtz-Zentrum Geesthacht, 21502 Geesthacht, Germany; wenyan.zhang@hzg.de
[4]   Ocean College, Zhejiang University, Zhoushan 316021, China
[5]   Physical Oceanography Laboratory, Ocean University of China, Qingdao 266100, China; sxj@ouc.edu.cn
[6]   Qingdao National Laboratory for Marine Science and Technology, Qingdao 266237, China
[7]   Institute for Advanced Ocean Study, Ocean University of China, Qingdao 266100, China; yingzhou@ouc.edu.cn
[8]   College of Oceanography, Hohai University, Nanjing 210098, China
*   Correspondence: zhanghan@sio.org.cn (H.Z.); zhoufeng@sio.org.cn (F.Z.)

**Abstract:** Surface waves induced by tropical cyclones (TCs) play an important role in the air–sea interaction, yet are seldom observed. In the 2017 summer, a wave glider in the northern South China Sea successfully acquired the surface wave parameters when three TCs (Hato, Pakhar, and Mawar) passed though successively. During the three TCs, surface wave period increased from 4–6 s to ~8–10 s and surface wave height increased from 0–1 m to 3–8 m. The number of wave crests observed in a time interval of 1024 s decreased from 100–150 to 60–75. The sea surface roughness, a key factor in determining the momentum transfer between air and sea, increased rapidly during Hato, Pakhar, and Mawar. Surface waves rotated clockwise (anti-clockwise) on the right (left) side of the TC track, and generally propagated to the right side of the local cyclonic tangential direction relative to the TC center. The azimuthal dependence of the wave propagation direction is close to sinusoidal in a region within 50–600 km. The intersection angle between surface wave direction and the local cyclonic tangential direction is generally smallest in the right-rear quadrant of the TC and tends to be largest in the left-rear quadrant. This new set of glider wave observational data proves to be useful for assessing wave forecast products and for improvements in corresponding parameterization schemes.

**Keywords:** surface wave; ocean surface roughness; tropical cyclone; typhoon; wave glider

## 1. Introduction

Tropical cyclones (TCs) are one of the most destructive natural hazards. TCs can generate strong surface waves induced by the associated winds. TC-induced surface waves can reach a height of more than 10 m [1–4], which may damage offshore platforms and vessels, erode coastlines, threaten the coastal area, and cause economic and human life losses [5]. The damage of TC-induced waves can be amplified when superposed with tides, leading to storm surges [6,7].

TC-induced surface waves are believed to significantly influence the condition of air–sea interface. In a traditional view, surface waves increase the sea surface roughness [8,9], which in turn reduce wind speeds [10]. The breaking of the surface waves during TCs causes large amounts of sea spray

droplets in whitecaps and whipping spume [11–13], progressively forming a 'slip' surface at the air–sea interface. Thus, the surface ocean drag coefficient tends to a saturation level, and decreases when wind speed at 10 m height exceeds 30–40 m/s [8,9,14–16]. TC-induced surface waves also contribute to a turbulent upper ocean and a deepening of mixed layer [1,17–19], which further leads to upper ocean sea surface cooling and subsurface warming during TCs [19].

Energetic surface waves generated by TCs also play a key role in promoting matter transport across continental shelves [20–22] and shaping of coastal morphology [23]. It has been found that the interaction between currents and cyclone-induced local surface waves and swells can lead to massive resuspension of sediments from the coastline till seafloor at several tens of meters water depth [22]. Sediments resuspended in a short period due to enhanced wave–current interaction during typhoons or storms, especially when local surface waves are aligned with swells, may form a thin (normally within 20 cm) but highly concentrated (>10 kg/m$^3$) layer in the immediate vicinity of the seafloor and transported offshore [20,22]. This process, together with diluted transport in the bottom boundary layer, are found to be the major mechanism for transporting solid material from continent to deep ocean [21]. Therefore, for a robust quantitative assessment of matter transport across continental shelves, which is the key part of the global source-to-sink transport pathway, it is imperative to accurately predict surface wave spectra and their dynamics generated by TCs.

The analyses of wind and wave measurements under TCs conditions demonstrate that the surface wave developments inside TCs follow the fetch- and duration-limited wave growth functions derived from steady wind forcing conditions [24–29]. The wind can pull the surface water along in the same direction because of surface friction, and surface water gains energy from the wind, then forms waves. For the fetch- and duration-limited wave system, two sets of parameters connect the wind–wave triplets, that is, wind parameter (i.e., wind speed at 10 m elevation) and wave parameters (i.e., significant wave height and spectral peak wave period). The wind speed at 10 m height within TCs is related to the TC intensity and can be estimated by the TC maximum wind speed (or central sea level pressure) and the radius of maximum wind speed. The wind fetch refers to the distance that how far the effective wind forcing on the formation of waves in a constant direction. A longer wind fetch will lead to more developed wind-induced waves. The TC-induced wave field hence is dependent on the TC properties, such as TC intensity, translation speed, and track [4]. For example, growth of surface waves will be largely hindered if their group velocity exceeds the TC translation speed, otherwise, they will get a longer fetch and the local wind-induced waves are able to interact with previously generated swells (causing nonlinear wave–wave interactions). In the northern (southern) hemisphere, TC-generated surface waves travel in the same direction as there exists an extended wind fetch on the right (left) side of the TC track.

Surface wave field inside TCs is complicated and multiple wave systems are frequently observed. TC-induced waves can have impacts very far from their tracks, reaching more than 10 times of the radius of maximum wind (RMW) [26,30,31]. Sea surface wave height is usually considered as a function of radial distance from the TC center by empirical relationships [32–34]. TC-induced wave spectra rapidly evolve and vary spatially by radius away from the center of the TC and the quadrant of the TC [3,25,35,36]. TC-induced waves are asymmetrical and the directional spectra possesses unique characteristics in each quadrant [31,37]. The directional wave spectra are often unimodal or bimodal, sometimes trimodal. In the right–forward quadrant of a TC where the highest waves are observed, the wave spectra are predominantly unimodal, while in the rear quadrants the wave spectra tend to be bimodal or trimodal [31,38]. Swells dominate the surface waves at the front of and outside the central typhoon region [39], and the dominant feature in the front of TC coverage area is single wave systems that propagate toward left and left–front [40]. Multiple wave systems are generally observed in the back and right quarters outside the RMW, and their directions and locations of occurrences are usually Gaussian distributed [40].

Although surface waves during TCs have been widely studied in the previous decades, direct in situ observations are still scarce. Owning to the hazardous sea conditions with high winds and large

waves, in situ measurement of surface waves within a TC is difficult. Most measurements are made by moored buoys [2,3,13,39] or remote sensing, including satellite and airborne synthetic aperture radar [41,42] and airborne scanning radar altimeter [38,43]. Actually, sea surface waves are important for TC remote sensing, especially for the synthetic aperture radar, since TC wind-induced sea surface waves can be "seen" by the microwave [44]. Recently, wave gliders have been used as a new platform to observe the air–sea interface during TCs [45,46]. In 2017, our wave glider "Black Pearl" (Ocean University of China, Qingdao, Shandong, China; Tsingtao Hydrotech Co., Ltd, Qingdao, Shandong, China) in the northern South China Sea successfully recorded surface waves during three TCs (Hato, Pakhar, and Mawar), providing a new set of data to study the TC-induced surface waves. This study aims to analyze the time series of wave data recorded by the glider in order to: 1) derive further insights into ocean surface wave response to TCs, and 2) assess existing wave forecast products for further improvement. The remainder of this paper is organized as follows: The observational/modeling methodology and the characteristics of the three TCs, are introduced in Section 2; the results of the observed sea surface waves and ocean surface roughness are shown in Section 3; the discussions of this work are presented in Section 4; and our conclusions are drawn in Section 5.

## 2. Data and Method

### 2.1. Wave Glider

Wave data were recorded by the "Black Pearl" wave glider which is shown in Figure 1b. The wave glider has a surfboard-like float at the surface and a sub with spring loaded paddles that uses wave energy for motion. It has solar panels on the float for supplying power to the control system, science instruments, and communications gear. It has a Global Positioning System (GPS) for precise location measurement and Iridium antennas for communications. Powered by wave and solar energy, the "Black Pearl" wave glider can operate individually or in fleets delivering real-time data for up to a year without requirement for additional fuel. See Li et al. (2017) [47] for more details of the "Black Pearl" wave glider.

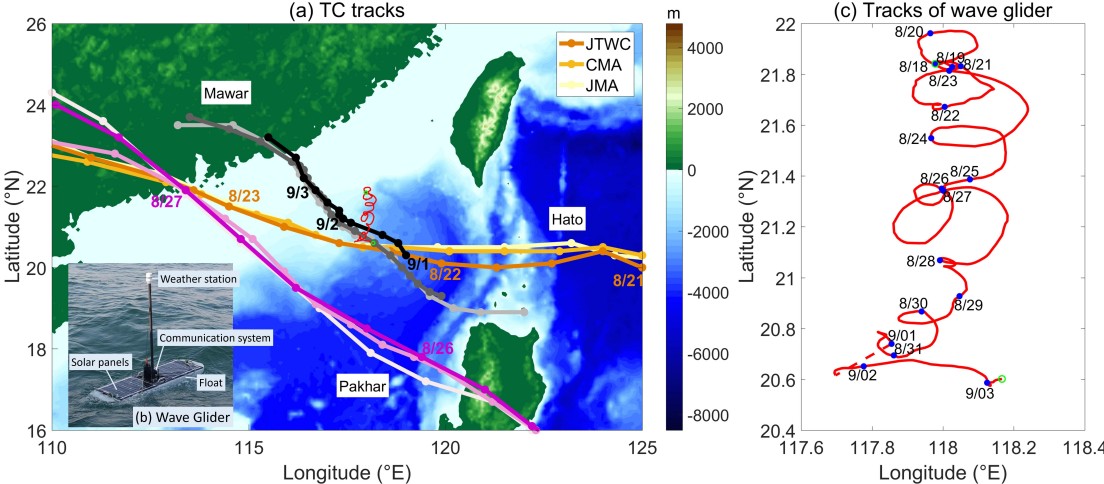

**Figure 1.** (**a**) Tracks of Hato (yellow colors), Pakhar (pink colors), and Mawar (black colors) from the Joint Typhoon Warning Center (JTWC; dark colors), China Meteorological Administration (CMA; middle colors), and Japan Meteorological Agency (JMA; light colors) data sets, normally showing their position every 6 h (dots). The CMA data set gives the position every 3 h when the tropical cyclone is near the shore. (**b**) Wave glider photo. (**c**) The positions of the wave glider are denoted by red line, with start–end positions marked with green circles (also shown in Figure 1a). The numbers are the observation dates.

The wave glider, equipped a SWS-1 wave gauge (Ocean University of China, Qingdao, Shandong, China; Tsingtao Hydrotech Co., Ltd, Qingdao, Shandong, China), was deployed in the South China Sea on 18 August 2017 and retrieved at 14:48 UTC (Universal Time Code) on 3 September 2017. The tracks of the wave glider are shown in Figure 1c. The valid range of wave height, wave period, and wave direction observed by the SWS-1 wave gauge was 0.2–30 m, 2–25 s and 0–360°, with an accuracy of 0.2 m plus the 5% measurement, 0.25 s and 5°, respectively. The observed data were transmitted via the Iridium satellite communication every 10 minutes. Wave data were recorded at a frequency of 2 Hz and output every 3 minutes with the average of 12 minutes. Following national marine standards and regulations, the number of wave crests is recorded in a time interval of 1024 s.

## 2.2. Tropical Cyclones

Three TCs, namely Hato, Pakhar, and Mawar, influenced our wave glider in the summer of 2017 successively (Figure 1a). The TC best-track datasets are from the Joint Typhoon Warning Center (JTWC [48]; www.usno.navy.mil/NOOC/nmfc-ph/RSS/jtwc/best_tracks), the China Meteorological Administration (CMA; http://tcdata.typhoon.org.cn/zjljsjj_zlhq.html; [49]), and the Japan Meteorological Agency (JMA; http://www.jma.go.jp/jma/jma-eng/jma-center/rsmc-hp-pub-eg/besttrack.html; [50]), respectively. The TC tracks from these three sources were consistent (Figure 1a), therefore we mainly used the JTWC data in this paper as JTWC data contains information of the TC's RMW. To be consistent, we interpolated the TC best-track data linearly to the series with the same temporal resolution as the observed wave data. Note that TC may not move along a straight line in a time interval of 6 h.

The basic characteristics of Hato, Pakhar, and Mawar are shown in Table 1. Hato formed in the northwest of the Pacific Ocean (127.0° E, 19.8° N) at 06 UTC 20 August, and became a tropical storm after 24 h. It grew into a typhoon when approaching the wave glider, with a translational speed of 6.65 m/s. The wave glider biased about 130.35 km to the right side of Hato's track (at about radius $7R_{max}$). After that, Hato travelled northwestward, and reached its peak intensity equivalent to a category-3 typhoon (based upon the maximum sustained wind speed) at 03 UTC 23 August (at about radius $24R_{max}$).

**Table 1.** Information of Hato, Pakhar, and Mawar when closest to the wave glider and when the tropical cyclone (TC) was strongest (in brackets) during the observation. Positive (negative) values of the distance to tropical cyclone track refers to the right (left) side of the track.

|  | Hato | Pakhar | Mawar |
|---|---|---|---|
| Distance to tropical cyclone track ($R$, km) | 130.347 (447.815) | 250.633 (471.136) | −49.441 (207.916) |
| Longitude (°E) | 117.802 (113.735) | 116.655 (113.446) | 117.704 (116.698) |
| Latitude (°N) | 20.523 (21.84) | 19.247 (21.843) | 21.061 (21.903) |
| Time | 8/22 09:41 (8/23 02:33) | 8/26 10:29 (8/26 23:43) | 9/1 17:13 (9/3 00:03) |
| Maximum wind speed ($V_{max}$, m/s) | 35.6 (50.7) | 22.5 (30.6) | 17.67 (23.1) |
| Minimum Air pressure ($P_{min}$, hPa) | 972 (949) | 994 (983) | 996 (989) |
| Translation speed ($U$, m/s) | 6.65 (8.80) | 10.14 (10.07) | 3.49 (2.11) |
| Radius of maximum wind ($R_{max}$, km) | 18.52 (18.52) | 46.3 (46.3) | 101.86 (92.6) |
| Non-dimensional TC translation speed, ($S = \frac{U}{R_{max}f}$)* | 7.02 (8.76) | 4.68 (4.12) | 0.71 (0.47) |

Pakhar was also located in the northwest of the Pacific Ocean (128.9° E, 15.7° N) when it formed at 00 UTC 24 August, and grew into a tropical storm 18 h later. Its minimum distance to the wave glider was about 250.6 km (at about radius $5R_{max}$) at 10 UTC 26 August, with a translation speed of 10.14 m/s. Its intensity grew while moving further away from the wave glider, and reached its greatest intensity at radius $10R_{max}$. After landing, Pakhar degraded to a tropical depression at 12 UTC 27 August.

Mawar formed in the South China Sea (119.0° E, 20.3° N) at 00 UTC 01 September. It moved northwestward slowly, growing from a low pressure into a tropical storm at 18 UTC 01 September. When approaching the wave glider, Mawar interrupted the communication of the wave glider at 00:13 UTC 01 September. The recorded closest location for the wave glider was about 49.44 km to the left side of Mawar track (at about radius $0.5R_{max}$), at a translational speed of 3.49 m/s. The communication re-established at 17:13 UTC 01 September. Mawar intensity peaked at about $2R_{max}$. After landfall,

Mawar degraded to a tropical depression at 18 UTC 03 September. Usually, the TC eye decreases in size as it strengthens, in other words, a larger RMW is normally found in a weaker TC. Therefore, compared to Hato and Pakhar, the weaker TC Mawar had a larger RMW.

The relative position of the wave glider in the moving coordinated system with the origin at the typhoon center is shown in Figure 2a. When effectively influenced by Hato and Pakhar, the wave glider was located on the right side of their tracks. During Mawar, the wave glider was located directly on its track and close to its center. The distance from the center of each TC to the wave glider, the TC's maximum sustained wind speed, and the minimum sea level pressure are shown in Figure 2b. The three TCs all peaked their intensity when moving away from the wave glider.

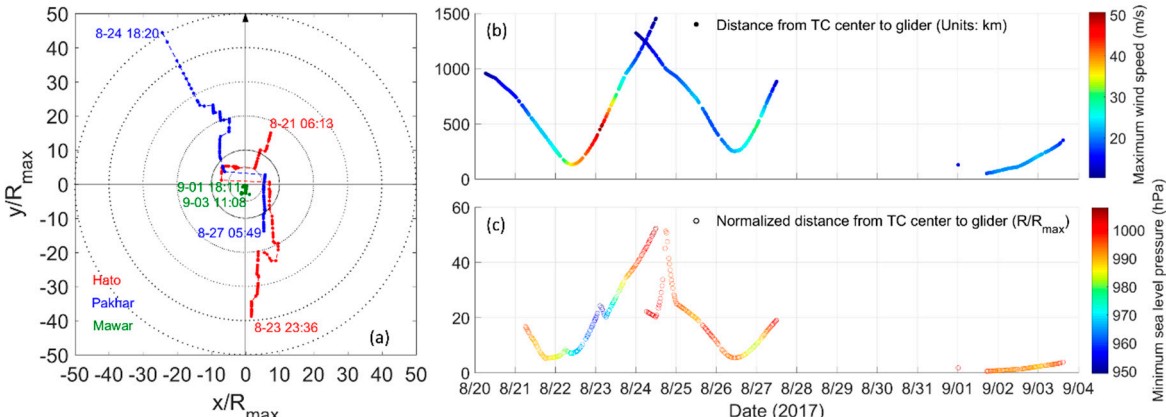

**Figure 2.** (**a**) Relative position of the wave glider in the moving coordinated system with the origin at the typhoon center and the y-axis in the typhoon's propagation direction (tropical cyclone (TC) wind speed larger than 17.2 m/s). The coordinates are scaled by the maximum wind speed radius $R_{max}$; (**b**) Distance from TC center to the wave glider (units: km). The color depicts the TC maximum wind speed (units: m/s) at radius $R_{max}$. (**c**) Normalized distance from TC center to the wave glider (units: 1) scaled by the radius of the maximum wind speed. The color depicts the TC minimum sea level pressure (units: hPa).

### 2.3. Ocean Surface Roughness

Since our wave glider did not observe wind speeds, ocean surface roughness in this study was calculated from specified wave properties following the relationship in Equation (1) given by Taylor and Yelland (2011) [51]. This relationship is incorporated in the version 3.0 of COARE (Coupled Ocean–Atmosphere Response Experiment) model described in Fairall et al. (2003) [52]:

$$Z_o = 1200 h_s \left( h_s / L_p \right)^{4.5} \tag{1}$$

$$L_p = \frac{g T_p^2}{2\pi} \tag{2}$$

where $Z_o$ is surface roughness (unit: m), $h_s$ is the significant wave height (unit: m), and $L_p$ is the peak wavelength (unit: m). $L_p$ is associated with the peak wave period $T_p$ (unit: s) and can be computed by using the standard deep-water gravity wave relationship in Equation (2). g is the acceleration due to gravity, and $h_s / L_p$ represents the slope of dominant waves. In the above equations, the coefficients are empirical and are based on fits to extensive observations [50,51].

### 2.4. Ideal Wind Model

One simple wind model named the SLOSH (Sea, Lake and Overland Surge from Hurricanes) model developed by the National Weather Service, National Oceanic and Atmospheric Administration, USA [53] was used to construct the wind speeds at observation locations. The SLOSH model defines the wind speed as follows:

$$V = V_{max} \frac{2R_{max}R}{R_{max}^2 + R^2} \tag{3}$$

where $R_{max}$ is the radius of maximum wind, $R$ is the distance from the wave glider to TC center, $V_{max}$ is the maximum wind speed at $R_{max}$, and $V$ is the wind speed at distance $R$.

Then $V$ is adjusted to the standard 10 m height by the following equation:

$$V_{10} = K_c K_m V \tag{4}$$

where $K_c$ and $K_m$ are correction factors and empirical. Following the work of Zhang et al. (2016) [2], we use $K_c = 0.92$ and $K_m = 0.8$ in this study.

## 3. Results

### 3.1. Surface Waves During Hato, Pakhar, and Mawar

The wave glider moved southward with a meandering track showing near-inertial oscillations, suggesting a dominance of near-inertial currents during observation.

TCs mainly induce waves with large wave height and long wave period. Affected by Hato, Pakhar, and Mawar, both wave height and wave period showed three peaks (Figure 3). The correlation between wave height and wave period was 0.79. Background mean wave height was about 0.41 ± 0.07 m and the wave period was about 5.8 ± 0.69 s before 18 UTC 20 August (i.e., before arrival of the first typhoon Hato). When Hato approached its nearest location to the wave glider, significant wave height peaked at 6.21 m and wave period peaked at 9.85 s rapidly, indicating that the wave field was dominated by Hato. Significant wave height (wave period) then gradually decreased to about 1.1 m (7.0 s) at 00 UTC 25 August when Hato moved away. About 38 h later, significant wave height (period) rose again to 3.56 m (8.0 s) at 14 UTC 26 August, due to the arrival of Pakhar, and afterward decreased to 0.97 m (6.2 s) at 00 UTC 29 August. Later on, Mawar increased the wave height (period) to 5.11 m (8.4 s) at 10:45 UTC 02 September. Different with the situation in Hato during which highest and longest waves were recorded when the TC center approached its nearest location to the wave glider, significant wave height and period during Pakhar and Mawar peaked between the time when the TC center was nearest to the wave glider and the time when the TC intensity was strongest.

There were three troughs for the number of wave crests in a time interval of 1024 s during the TC period, corresponding to the three peaks of the wave height and wave period. The correlation coefficient between wave crest number and wave period was −0.94. The temporal variations of significant wave height, mean wave height, 1/10 mean wave height, peak wave height, and spectral significant wave height were consistent, all affected by both TC intensity and the relative distance to TC (Figure 4).

Waves rotated clockwise (anti-clockwise) on the right (left) side of the TC tracks (Figure 3d), which is consistent with previous research [2] (see Figure 5 for wave vectors in the longitude–latitude coordinates). To further analyze the variation of waves during the three tropical cyclones, the wave vectors in a reference grid that moved with TC (Figure 6) were plotted. The pattern of wave vectors observed by the wave glider (Figure 6) suggests that dominant waves radiated out from a region with intense TC wind, i.e., around the RMW. During Hato and Pakhar, the wave glider was located mainly on the right side of the TC tracks and recorded anticyclonically rotated waves correspondingly. During Mawar, the wave glider was located on the left side of the TC track and quite close to the TC center, and the recorded waves mainly rotated cyclonically.

Consistent with previous works [28,40], this study also indicates that variations of the wave propagation directions are primarily azimuth dependent and their dependence on radial distance from the TC center is relatively weak (Figure 7). In other words, wave direction is significantly affected by the azimuth angle of the observation location referred to the TC heading, while slightly affected by the radial distance from the TC center. As shown in Figure 7, wave direction ($\varnothing_w$, in degree) measured from the TC heading was related to the azimuth angle of the observation location $\varnothing$. Generally, the angle of wave direction ($\Delta\varnothing_w$), relative to the location vector normal ($\varnothing_{\vec{n}}$), was in the sinusoidal curve variation with the azimuth angle of observation location $\varnothing$. Herein, the location vector normal ($\varnothing_{\vec{n}}$), i.e., location cyclonic tangential direction relative to TC center, represents the ideal local wind direction of the TC. Therefore, $\Delta\varnothing_w$ depicts the angle bias of surface wave propagation direction to the ideal local wind direction. The negative $\Delta\varnothing_w$ suggests that the surface waves generally propagate to the right side of the cyclonic tangential direction relative to TC center. In the left quadrant of the TC, $\Delta\varnothing_w$ generally ranges from −90° to −180°; while in the right quadrant, the range of $\Delta\varnothing_w$ is mainly from −45° to 45°. The radial dependency of $\Delta\varnothing_w$ is characterized by a nearly constant offset of the sinusoidal curves, except for the locations far away from the TC center. Note that previous understanding of the radial dependence and azimuthal dependence of wave propagation direction is derived from the data in the TC region within 50–200 km [28,40], and can be applied to a much larger region up to 600 km according to our results.

Since $\Delta\varnothing_w$ depicts the angle bias of surface wave propagation direction to the ideal local wind direction, $\Delta\varnothing_w$ can also represent the alignment between surface wave propagation direction and wind direction. As shown in Figures 6 and 7, $\Delta\varnothing_w$ is generally smallest in the right–rear quadrant, followed by the right–forward and the left–forward quadrant, and largest in the left–rear quadrant. This suggests that the TC-induced wave direction is more closely aligned with the local wind direction in the right–rear quadrant with respect to the forward direction, except for those close to the TC track. That is because when close to the TC track, the wave glider may swing back and forth on the right and left sides of TC. The response of local wind driven waves is much slower than the changing rate of the wind direction, leading to an imperfect alignment between surface waves and wind. The result of surface wave and wind alignment is supported by the finding of Hu and Chen (2011) [37], who demonstrated that wave spectra in the right–rear quadrant are mainly controlled by the local strong winds, while in other quadrants are dominated by swells originated in distant regions; however, the study of Hu and Chen (2011) only considered the wave field within $8R_{max}$ from the TC center, which is much smaller than the impact range in this study, e.g., more than $25R_{max}$ for Hato and $15R_{max}$ for Pakhar.

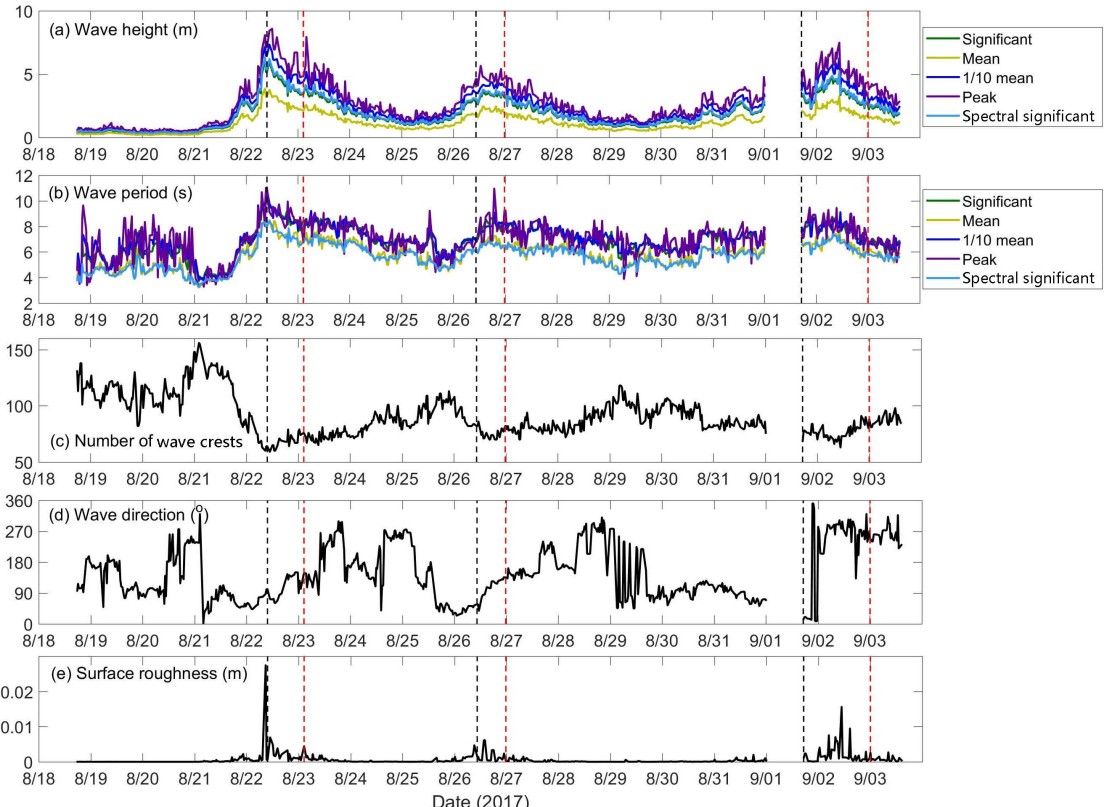

**Figure 3.** Wave parameters from the wave glider observations: (**a**) wave height; (**b**) wave period; (**c**) number of wave crests in a time interval of 1024 s; (**d**) wave direction; and (**e**) ocean surface roughness. The black dashed line represents the time when the TC is closest to the wave glider. The red dashed line shows the time when the TC is strongest during the observation.

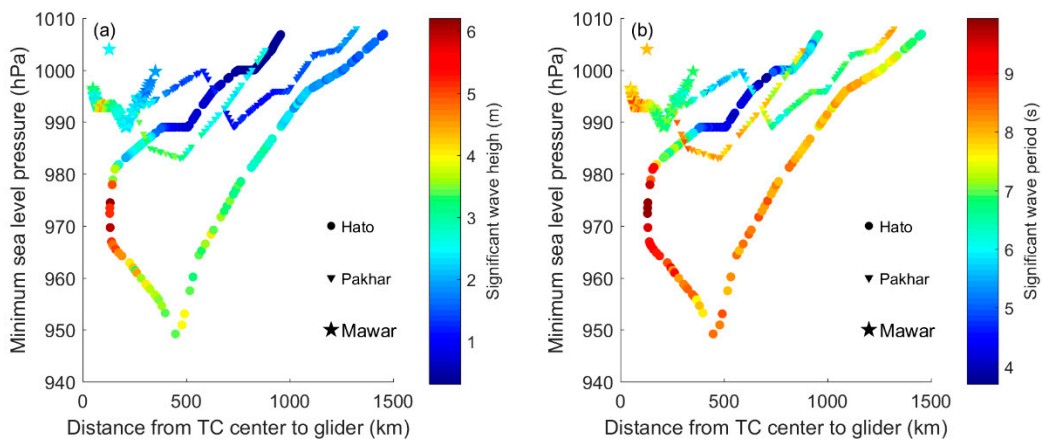

**Figure 4.** Dependency of wave parameters on TC intensity and the distance from TC center to glider: (**a**) significant wave height (color coding); (**b**) significant wave period (color coding).

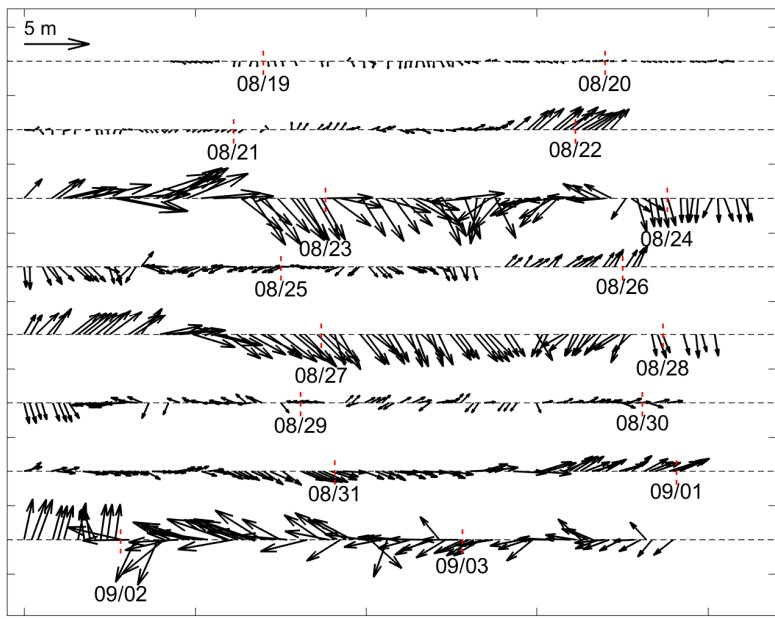

**Figure 5.** Vector direction indicating the mean wave direction during the observed period (numbers depicts the observation date). The length of the vector depicts the significant wave height. The northward direction is toward the top of the page.

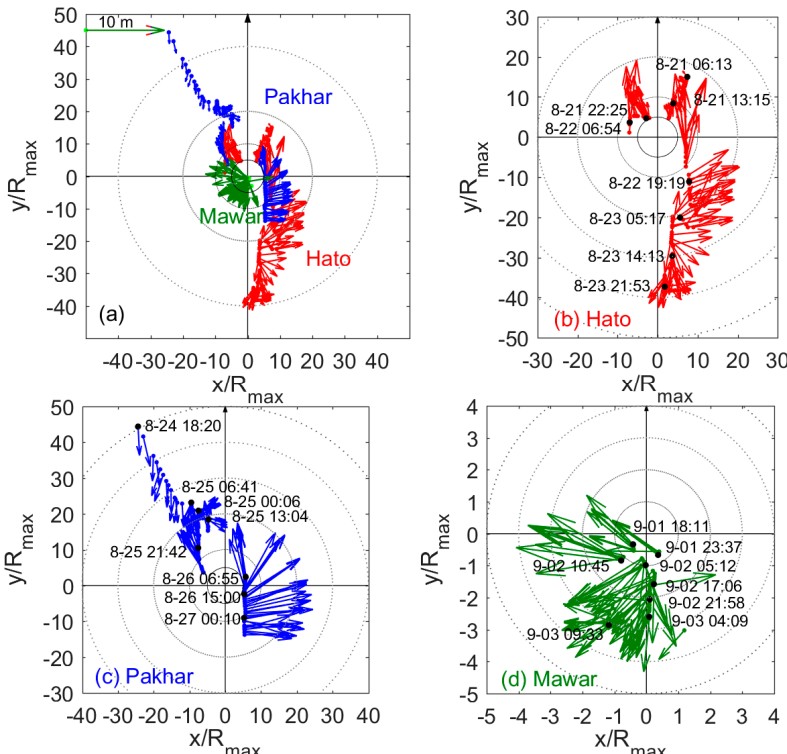

**Figure 6.** The vector directions indicating the mean wave directions for the wave glider data during the three typhoons (only when TC wind speed larger than 17.2 m/s). The vector length indicates the significant wave height. Wave glider positions are indicated by dots. All data are plotted in the moving coordinated system with the origin at the typhoon center and the y-axis in the typhoon's propagation direction. The dashed circles represent the different times of the maximum wind speed radius ($R_{max}$). The system is shown in the Northern Hemisphere (i.e., anti-clockwise circulation) (**a**–**d**).

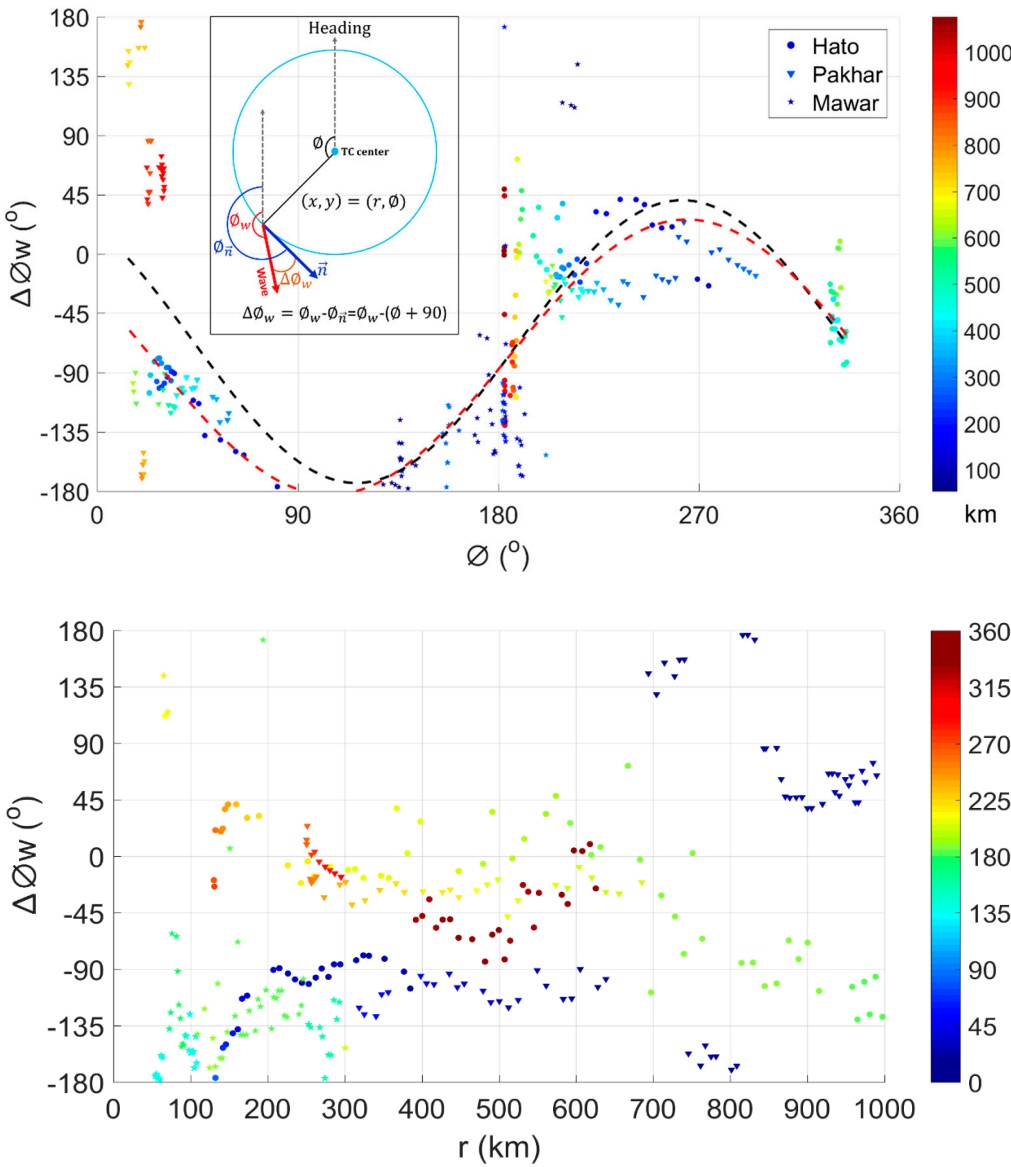

**Figure 7.** Dependency of the wave direction on azimuthal variation and radial distance for Hato (dot), Pakhar (triangle), and Mawar (pentagram). $\varnothing$ is the azimuth angle of the observation location; $\varnothing_w$ represents the wave direction measured from the TC heading; $\varnothing_{\vec{n}}$ represents the angle of the location vector normal representing the ideal local wind direction; and $\Delta\varnothing_w$ represents the angle of wave direction relative to the location vector normal. Negative (positive) $\Delta\varnothing_w$ suggests the TC wave propagates to the right (left) side of the local wind. The black smooth dashed line is one-term-Fourier-fit curve through all the data and the red dashed line is one-term-Fourier-fit curve through the data within 700 km radial distance. The color coding represents the radial distance from the TC center (top) and azimuth angle (bottom), respectively.

### 3.2. Ocean Surface Roughness

Ocean surface roughness, related to the wind stress, is important for the momentum transfer at the air–sea interface. For wind speeds larger than 5 m/s, surface waves are a dominant factor for ocean surface roughness. Ocean surface roughness has multi length scales, ranging from millimeter to meter [54]. The ambient ocean surface roughness primarily induced by turbulence and swell ranges from 0 to 0.015 m with an average level of 0.008 m from altimeter data [55]. Laboratory extreme wind experiments suggest a saturated ocean surface roughness length of about 0.003 m beyond 10 m height wind speeds of 33 m/s [8].

Consistent with the variation of wave height, the estimated surface roughness shown in Figure 3e presents three peaks of 0.028 m, 0.006 m, and 0.016 m, corresponding to Hato, Pakhar, and Mawar, respectively. The background ocean surface roughness was ~0 m. The rapid response of ocean surface roughness to TCs suggests strong dynamic processes occurring at the air–sea interface when high waves are induced by strong wind.

## 4. Discussion

### 4.1. Assessment for Wave Modelling

We compared our wave glider data with a set of 3-hourly-instantaneous analysis and forecast products for the global ocean surface waves (Figure 8). The wave products were from the global ocean analysis and forecast system of Météo-France, which is based on the Meteo France WAve Model (MFWAM), Copernicus Marine Environment Monitoring Service (CMEMS). The MFWAM model with a horizontal resolution of 1/12 degree is forced by Integrated Forecasting System-European Centre for Medium-Range Weather Forecasts (IFS-ECMWF) winds and uses the assimilation of altimeters. The wave products can be downloaded from the website of E.U. Copernicus Marine Service Information, CMEMS [56]. Figure 8 indicates that the model performance varies among different wave parameters when compared to wave glider observations. Temporal variation of spectral significant wave height was satisfactorily reproduced by the model with a root mean square error (RMSE) of 0.39 m and a mean relative bias of 5.6%, which are comparable with the observation accuracy of 0.2 m plus 5% measurement. The wave peak period was slightly overestimated (a RMSE of 1.60 s and a mean relative bias of 16.9%) in the model and the model-observation difference exceed the corresponding observation accuracy (i.e., 0.25 s). The modeled variation of the mean wave direction variation was significantly smaller than the glider observation with an accuracy of 5°.

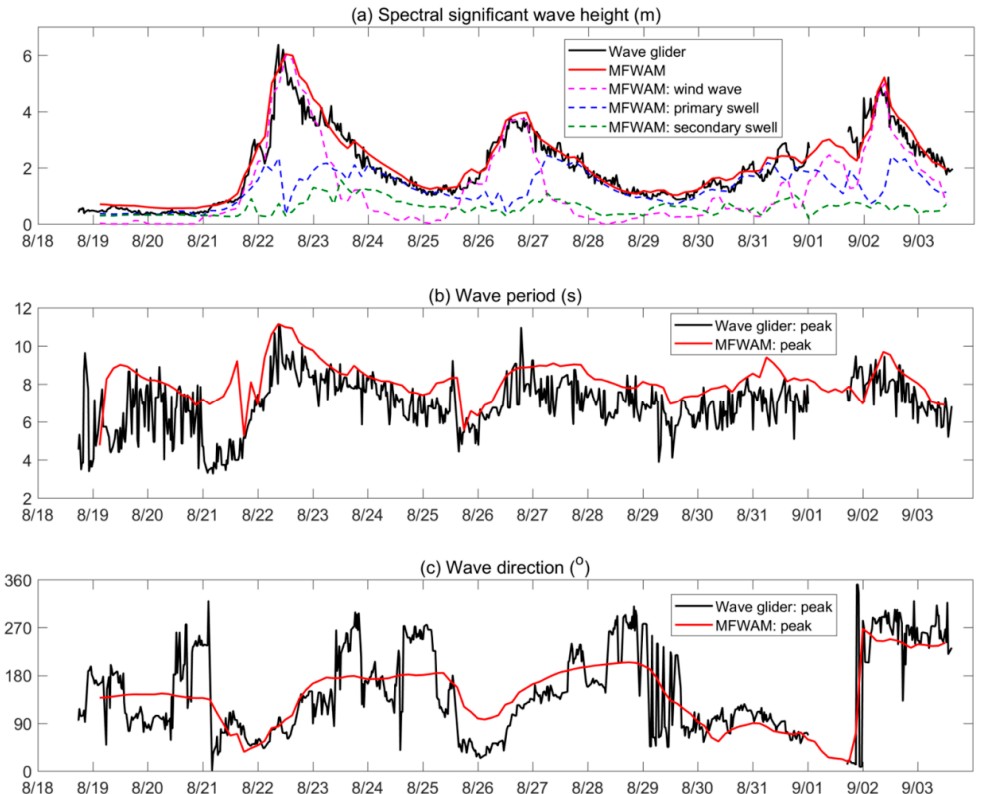

**Figure 8.** Comparison of wave glider observation and forecast product from the Meteo France WAve Model (MFWAM) (**a**–**c**).

The MFWAM model had a good performance in capturing the major spatial characteristics of wave height (Figure 9). Wave height in cases of Hato and Pakhar was significantly larger on the right side of TC due to the combined influence of the asymmetry of TC wind and the longer wind fetch, which is consistent with pervious views, while there was no obvious difference for wave height on two sides of Mawar. The MFWAM simulations are also in agreement with our wave glider observations (Figure 6), although one glider can hardly capture complete spatial characteristics of surface waves. Note that the MFWAM model divided the wave height into different components related to swells and winds, respectively. We can see that local wind driven surface waves dominated during the TC periods, i.e., approximately 22–23 August for Hato, 26–27 August for Pakhar, and 02–03 September for Mawar (Figure 8a). The wind wave height decreased rapidly when TC moved far away, and the surface wave height was mainly controlled by swells.

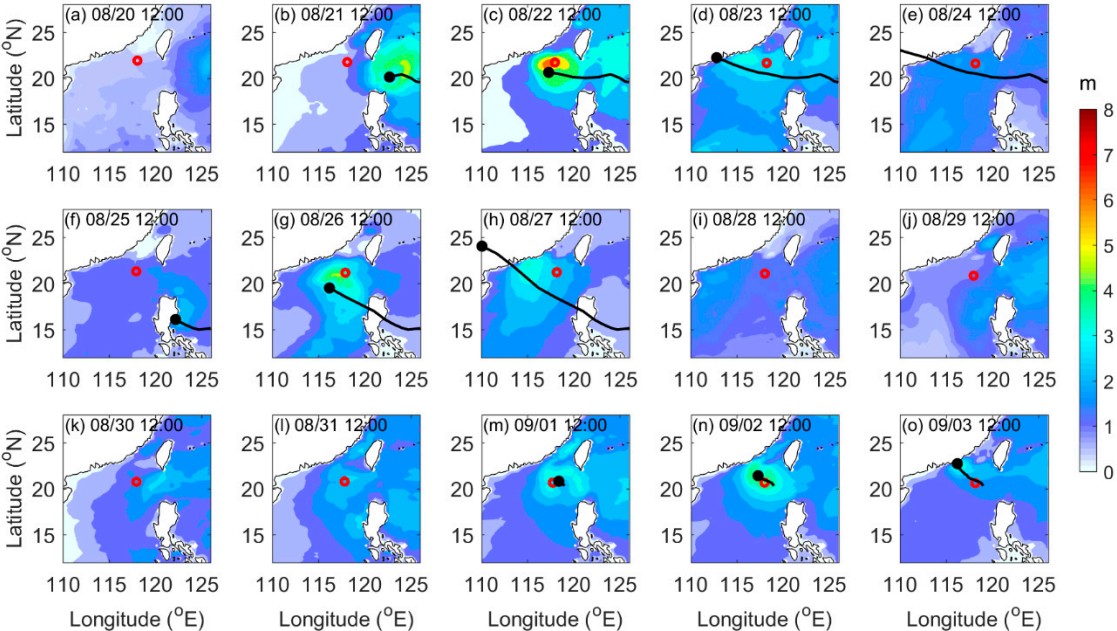

**Figure 9.** Spectral significant wave height (unit: m) simulated by global wave model MFWAM at 12:00 UTC (Universal Time Code) between August 20 and September 03, 2017. The red hollowed dots depict the position of the wave glider, the black lines depict the tropical cyclone tracks, and the black dots depict the centers of the tropical cyclones (**a**–**o**).

We also assessed the parameterization scheme of significant wave height for open oceans developed by Wang et al. (2017) [57] based on 15 years of hourly observational wind–wave data from eight buoys off the northwest coast of the United States. Two sets of data for wind at 10 m height were used (Figure 10a) as the wave glider did not observe wind. One was extracted from the Cross-Calibrated Multi-Platform (CCMP) Analysis Product with a horizontal resolution of 0.25° (produced by Remote Sensing Systems, USA), and the other one was reconstructed by the SLOSH scheme [2,52]. Figure 10b shows that the results of Wang et al. (2017) and CCMP are consistent with our observation for Hato and Pakhar but significantly underestimates for Mawar; Wang et al. (2017) and SLOSH performs better for Mawar but significantly underestimates for Hato and Pakhar. We therefore argue that CCMP data are reliable for capturing the general wind distribution pattern during TCs, but underestimates the wind strength (by ~50% in the case of Mawar) around the typhoon track due to its relatively coarse horizontal resolution. While the SLOSH scheme performs quite well for the area close to the typhoon track (within $R_{max}$), it significantly underestimates wind strength far away from the TC center (e.g., distance more than $5R_{max}$).

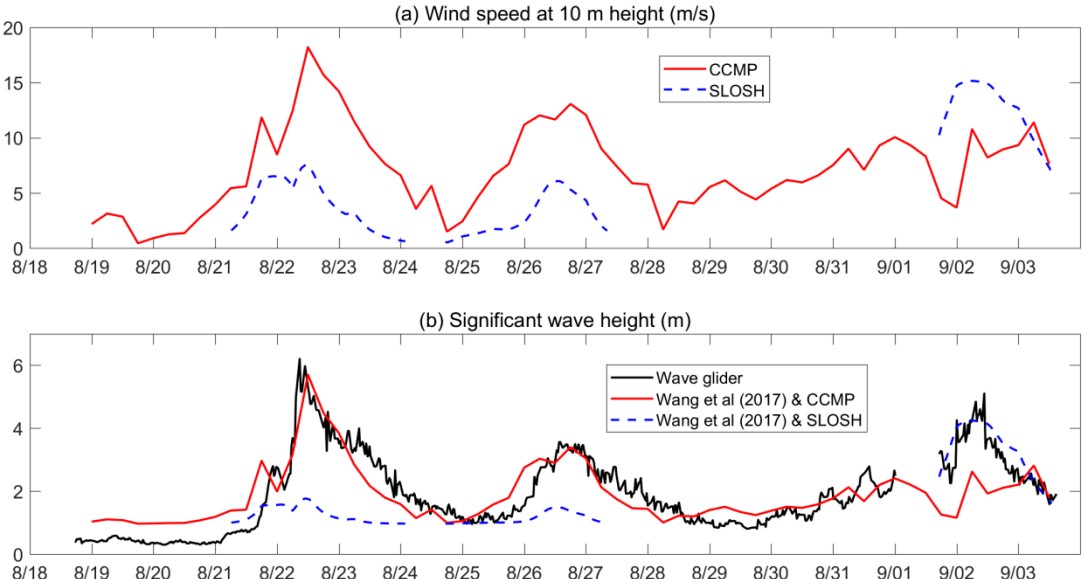

**Figure 10.** (**a**) Wind speed at 10 m height using Cross-Calibrated Multi-Platform (CCMP) data (solid red) and the Sea, Lake, and Overland Surges from Hurricanes (SLOSH) scheme (dash blue). (**b**) Significant wave height from the wave glider (solid black), the parameterization of Wang C. et al. (2017) [57] by using CCMP wind (solid red), and the parameterization of Wang C. et al. (2017) by using SLOSH wind (dash blue).

### 4.2. Ocean Surface Roughness

Since the CMEMS wave products are relatively reliable in capturing the general surface wave distribution pattern during TCs, we further analyzed the spatial pattern of ocean surface roughness induced by surface waves following the equations introduced in Section 2.3. TCs enhance the ocean surface roughness (mostly near the radius of maximum wind speed), which seems more obvious in the rear quadrants of TCs (Figure 11). Wave-induced surface roughness is associated with wave properties. In the right–rear quadrant where wave directions are closely aligned with the local wind direction, the surface waves tend to experience an extended wind fetch and grow larger, resulting in a more obvious increase of surface roughness. While in the left–rear quadrant, the peak wave frequencies are relatively high, that is probably related to the mixed sea states (comprised of wind sea and swell waves, with an angle between their propagation directions) in this region. Our wave glider observations also show a greater ocean surface roughness in the right–rear quadrant of TCs (Figure 12). By altering the drag or friction of sea surface, ocean surface roughness further affects the momentum and energy transfer across the air–sea interface, e.g., wind tends to input more energy in the rear quadrant of a TC, eventually affecting TC–ocean interactions and TC intensity. Therefore, surface wave spectra are essential to improve the prediction of TC intensity.

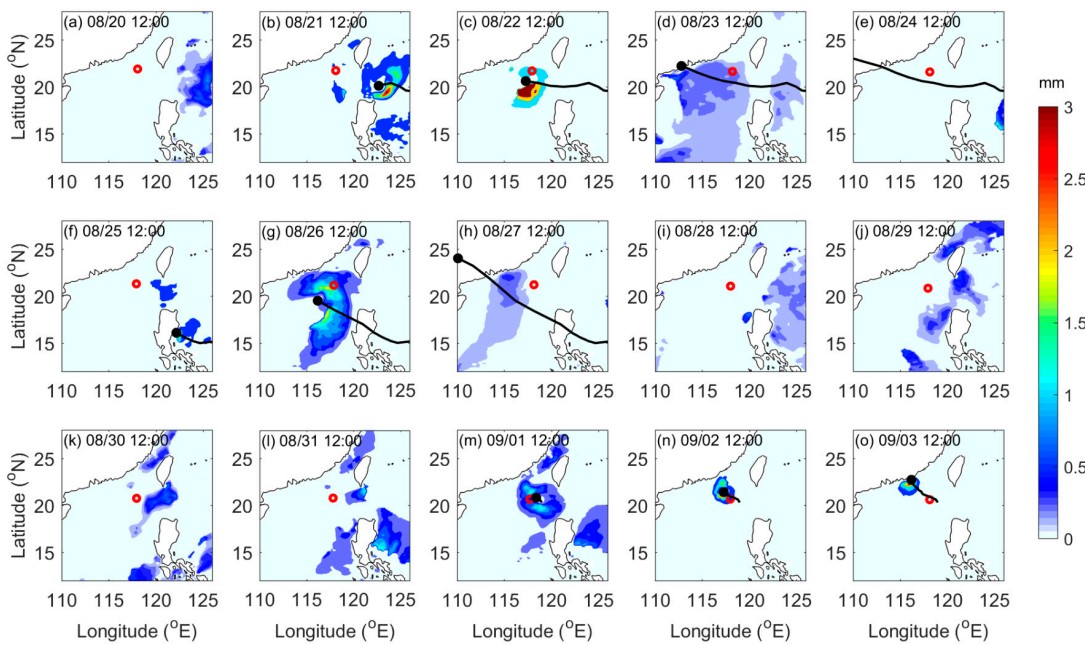

**Figure 11.** Ocean surface roughness (unit: mm) estimated by global wave model MFWAM products at 12:00 UTC between August 20 and September 03, 2017. The red hollowed dots depict the position of wave glider, the black lines depict the tropical cyclone tracks, and the black dots depict the centers of the tropical cyclones (**a**–**o**).

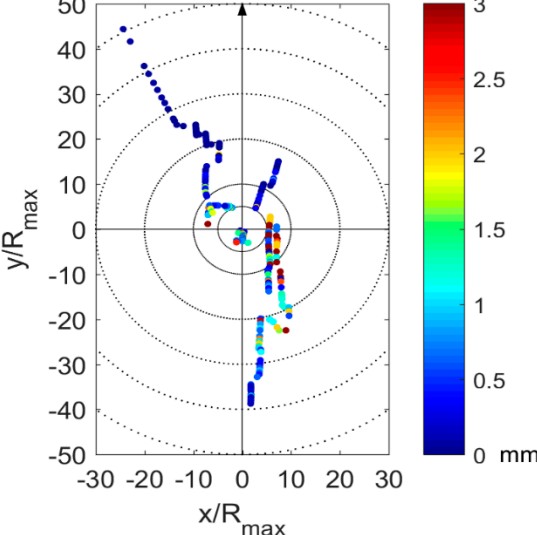

**Figure 12.** Ocean surface roughness (unit: mm) estimated by wave glider observational data in the moving coordinated system with the origin at the typhoon center and the y-axis in the typhoon's propagation direction (TC wind speed larger than 17.2 m/s). The coordinates are scaled by the maximum wind speed radius $R_{max}$.

### 4.3. Future Needs

To further understand the air–sea interaction during a TC, simultaneous in situ atmospheric and oceanic data are needed besides surface waves. Meteorology data (such as wind, pressure, humidity, and rainfall) and ocean data (such as sea surface temperature, salinity, and current) can be obtained by equipping the wave glider with corresponding sensors. Especially, simultaneous observation of surface winds and surface waves is essential to understand the wind–wave coupling and drag coefficient (or ocean surface roughness), as well as other observations, such as surface wind observed

by scatterometer or synthetic aperture radar. If possible, multiple groups of wave gliders deployed at the same time would be a great help for studying the spatial structure of TC–ocean interactions.

## 5. Conclusions

Ocean surface wave variations during three tropical cyclones, i.e., Hato, Pakhar, and Mawar, were investigated based on in situ measurement acquired by a wave glider deployed in the northern South China Sea during the summer of 2017. The state-of-the-art glider observations support the following conclusions.

The recorded responses of ocean surface wave parameters, such as wave height, wave period, and the number of wave crests, are strongly dependent on the TC intensity and the relative distance from glider to the TC center. During Hato, the observed surface wave peaked in size when the wave glider was closest to TC center. During Pakhar and Mawar, the largest surface waves were found between the time when the wave glider was closest to the TC center and when the TC reached its peak intensity. The rapid responses of ocean surface roughness during the three TCs suggests strong dynamic processes in the air–sea interface.

The surface wave direction was highly azimuthal dependent and was of weakly radial dependence. The observed surface waves rotated clockwise on the right side of TC track and anti-clockwise on the left side. Generally, surface waves propagated to the right side of the local cyclonic tangential direction relative to TC center, with an intersection angle varying sinusoidally in a region up to about 600 km, which is much further than we knew before. The intersection angle between surface wave direction and the local cyclonic tangential direction was generally smallest in the right–rear quadrant of a TC and tended to be largest in the left–rear quadrant, suggesting a closer alignment between wave direction and local wind direction in the right–rear quadrant.

Last but not least, this study demonstrated wave glider data are helpful for evaluation of wave forecast products and potential improvement of parameterization schemes. For example, the overall performance of MFWAM wave model in wave field, including wave height and wave period, is satisfactory, while its performance in wave direction needs to be improved. The current schemes (CCMP and SLOSH) for reconstruction of high-resolution wind fields during typhoons also need to be improved.

**Author Contributions:** H.Z. and F.Z. designed research; D.T. performed research and wrote the original manuscript; W.Z. improved the manuscript; X.S. and Y.Z. deployed and recovered the wave glider and provided details of the wave glider observational data; D.T., H.Z., W.Z., X.S., Y.Z., and D.K. All authors have read and agreed to the published version of the manuscript.

**Funding:** This work was funded by the National Key Research and Development Program of China (No. 2016YFC1401603 and 2018YFC1506403), the Scientific Research Fund of the Second Institute of Oceanography, Ministry of Natural Resources (No. QNYC2002), the National Natural Science Foundation of China (No. 41705048, 41806021, 41621064, and 41976007), the National Program on Global Change and Air–Sea Interaction (No. GASI-IPOVAI-04), the China Ocean Mineral Resources Research and Development Association Program (No. DY135-E2-3-01), the National Program on Global Change and Air–Sea Interaction (the Joint Advanced Marine and Ecological Studies in the Bay of Bengal and the eastern equatorial Indian Ocean, JAMES), and the Senior Visiting "Ocean Star" Scholarship program provided by the State Key Lab of Satellite Ocean Environment Dynamics (SOED), Second Institute of Oceanography, Ministry of Natural Resources (No. QNHX1819).

**Acknowledgments:** We acknowledge the Joint Typhoon Warning Center (JTWC, http://www.usno.navy.mil/JTWC), the China Meteorological Administration (CMA, http://tcdata.typhoon.org.cn) and the Japan Meteorological Agency (JMA, http://www.jma.go.jp/jma/jma-eng/jma-center/rsmc-hp-pub-eg/besttrack.html) for providing the tropical cyclone tracks, the Copernicus Marine Environment Monitoring Service (CMEMS, http://marine.copernicus.eu) for providing the global wave forecast model product, and the Remote Sensing Systems (http://www.remss.com) for providing the CCMP Version-2.0 vector wind.

**Conflicts of Interest:** The authors declare no conflict of interest.

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
