# Peer review of "Wave Glider Observations of Surface Waves During Three Tropical Cyclones in the South China Sea"

_water, doi:10.3390/w12051331_

Round 1
Reviewer 1 Report
Please check the attachment.

Reviewer 2 Report
This paper provides quite significant results obtained in the ocean under tropical cyclones. Many interesting results are shown in the paper and I think it is suitable for a journal paper. However, there are some suggestions to improve the paper.
1) Image of wave glider may be introduced in the paper so that readers can understand the instrument employed in this work.
2) There are research of the waves under the cyclone. Authors are suggested to stress the new findings in this work.
3) It is not easy to distinguish the markers in some figures. There may be a room for improvement.
Reviewer 3 Report
Please see the attached file.

Round 2
Reviewer 1 Report
The authors have addressed all my concerns.